# Quantum Exact Response Theory Based on the Dissipation Function

**DOI:** 10.3390/e27050527

**Published:** 2025-05-15

**Authors:** Enrico Greppi, Lamberto Rondoni

**Affiliations:** 1Dipartimento di Scienze Matematiche, Politecnico di Torino, Corso Duca Degli Abruzzi 24, 10129 Torino, Italy; 2Istituto Nazionale di Fisica Nucleare, Sezione di Torino, Via Pietro Giuria 1, 10125 Torino, Italy

**Keywords:** quantum response theory, qubits, Lindblad equations

## Abstract

The exact response theory based on the Dissipation Function applies to general dynamical systems and has yielded excellent results in various applications. In this article, we propose a method to apply it to quantum mechanics. In many quantum systems, it has not yet been possible to overcome the perturbative approach, and the most developed theory is the linear one. Extensions of the exact response theory developed in the field of nonequilibrium molecular dynamics could prove useful in quantum mechanics, as perturbations of small systems or far-from-equilibrium states cannot always be taken as small perturbations. Here, we introduce a quantum analogue of the classical Dissipation Function. We then derive a quantum expression for the exact calculation of time-dependent expectation values of observables, in a form analogous to that of the classical theory. We restrict our analysis to finite-dimensional Hilbert spaces, for the sake of simplicity, and we apply our method to specific examples, like qubit systems, for which exact results can be obtained by standard techniques. This way, we prove the consistency of our approach with the existing methods, where they apply. Although not required for open systems, we propose a self-adjoint version of our Dissipation Operator, obtaining a second equivalent expression of response, where the contribution of an anti-self-adjoint operator appears. We conclude by using new formalism to solve the Lindblad equations, obtaining exact results for a specific case of qubit decoherence, and suggesting possible future developments of this work.

## 1. Introduction

Statistical physics provides a successful description of systems in thermodynamic equilibrium through probability distributions known as ensembles. However, most systems of interest are not in equilibrium. Response theory studies their behavior when subject to external actions, usually adopting a perturbative approach that received substantial attention in the 1950s with the works of Callen, Green, Kubo and others [1,2,3,4,5,6]. The perturbation is assumed to be small compared to the Hamiltonian and can be time-dependent. In classical mechanics, paralleled by quantum mechanics, the initial unperturbed state of the system is represented by a probability distribution in phase space, which is postulated to be affected by the external action like a fluid in real space is affected by external forces but preserves its mass. Therefore, a continuity equation for the probability, known as the Liouville equation, is taken to hold in phase space [7]. If the external action is small, one assumes that the effect on the probability density can be approximated to the first order in the magnitude of the action, using the time correlation function of the perturbation and the observable of interest, computed with respect to the equilibrium probability. Kubo provided a single formalism to treat both classical and quantum dynamics, concerning probability densities in the first case and density matrices in the second, that, in the linear approximation, yields susceptibilities for systems not far from the thermodynamic equilibrium. The need to develop suitable techniques in the quantum branch of this field is ever more important [8,9,10]. For instance, the phenomenon of decoherence, typical of open quantum systems, represents a major challenge in quantum computing [11,12]. A deeper understanding of quantum dissipative dynamics could lead to significant advances in several fields. In this regard, Kubo’s theory has recently been generalized to nonequilibrium situations described by general time-local master equations [13,14,15], such as Lindblad equations, which are the most widely used for describing open quantum dynamics. Thermal equilibrium states are replaced by nonequilibrium steady states, and dissipative perturbations are considered in addition to the Hamiltonian ones [16]. Extensions of linear response theory have been developed to also account for non-Markovian effects [17]. Different formulations of the response function have enriched the theory [18]. While these formulations produce equivalent results in numerous applications, the existence of different types of response functions provides important theoretical and experimental advantages, allowing one to choose the best one depending on the specific application.

As a result, this perturbative approach has achieved broad applicability and a considerable degree of completeness [19,20]. However, nonlinearities are common [21,22], especially when dissipation, phase transitions, and decoherence phenomena occur. In these situations, even a small perturbation can lead to significant modifications of the state, impairing the applicability of linear response. One possible approach to these issues consists in taking higher-order terms in the perturbation expansions, which may provide greater precision, but it is often cumbersome and expensive, and may still fail to capture phenomena like phase transitions and anomalous behaviors. More generally, the perturbations may not be small. In all such cases, the best solution would be an exact theory of response. In recent years, research aiming at this goal has started to emerge. One interesting approach is the use of stochastic methods from classical physics to study quantum dynamics [23,24]. The stochastic reset technique is particularly suitable for representing measurement processes and, in this context, for describing open quantum systems exactly [25]. While promising, this method has the disadvantage of being applicable only to a narrow range of quantum systems, and it is not always possible to move to “classical” stochastic dynamics.

In contrast, an exact response theory already exists in classical statistical mechanics, which originated from fluctuation theorems [6,26,27,28,29], a generalization of the second law of thermodynamics for small systems, allowing the dissipation to become a random variable that can take negative values. A quite general classical exact response theory has been built on the basis of the transient time correlation function (TTCF) [7] and of the fluctuation relations. The key ingredient of this theory is the Dissipation Function, first defined by Evans and Searles as the physically relevant quantity that verifies the fluctuation relation [30,31,32]. Its connection with exact response theory is referred to as the dissipation theorem in Ref. [33]; it concerns systems subjected to time-independent perturbations, such as an external constant field. A dynamical system interpretation has been provided [34]. This theory has offered numerous advantages and new perspectives, and it has been applied in several contexts [35,36]. It has allowed for treating hard nonequilibrium problems, at low drivings [37,38], drastically improving the signal-to-noise ratio, and providing a superior method with respect to direct averaging for such calculations. In recent applications, it was demonstrated that this method dramatically improves the accuracy of the results at low shear rates, and that it is suitable to investigate atomistically detailed confined fluids at realistic flow rates [37]. It has also been shown that the TTCF can be used to define a local diffusion coefficient, leading to important practical implications for nanoscale and inhomogeneous systems [39]. More recently, the Dissipation Function has been applied to polar molecules in an electric field, yielding excellent results [40]. Machine learning techniques have further advanced the study of the Dissipation Function in nonequilibrium steady states, leading to a more accurate, short-time valid steady-state fluctuation theorem [41]. Moreover, this exact response theory has recently been extended to more complex perturbations, such as stochastic ones [36]. The Dissipation Function and related response theory remain subjects of active research, and further interesting developments are expected in various fields.

Attempts to link quantum response theory with quantum fluctuation theorems have been explored but without advancing beyond perturbative approaches [42]. Therefore, in the present paper, we propose an exact quantum response theory based on the Dissipation Function. In Section 2, we review the fundamental aspects of the classical exact response theory. We formulate a quantum analogue of the Dissipation Function, introducing possible definitions for the corresponding Dissipation Operators. Finally, we derive two exact expressions for the computation of the observables expectation values, based on the new Dissipation Operator, analogously to the corresponding classical response theory. In Section 5, we apply the new expressions to qubit systems, where exact results can be obtained in other ways as well. We then compare the new approach with linear response theory in an application to a spin-1/2 particle in a magnetic field. This example highlights the advantages that an exact response theory can offer. In Section 6, we conduct a study of the Dissipation Operator, providing some results that could be useful in future research. In Section 7, we extend the new method to Lindblad dynamics, and apply it to an open qubit system characterized by decoherence. Under appropriate assumptions, the new expression based on the Dissipation Operator gives exact results for this particular case. In Section 8, we discuss the obtained results and suggest future developments.

## 2. Classical and Quantum Response Theory

In this section, we review the fundamental aspects of the classical Dissipation Function and its use in response theory [34]. We then propose a formulation for quantum dynamics.

Let us consider a system whose microscopic phase Γ∈M evolves according to the equation of motion $\dot{\Gamma }=G\left(\Gamma \right)$. We define the map $S^{t}:M\to M$. $S^{t}\Gamma$ is the solution at time *t* of the system with initial condition Γ∈M. We assume that the phase space M is endowed with a probability measure $d\mu_{0}\left(\Gamma \right)=f_{0}\left(\Gamma \right)d\Gamma$, of density $f_{0}$, which evolves according to the generalized Liouville equation $\partial_{t}f_{t}\left(\Gamma \right)=-\nabla_{\Gamma }\cdot \left(f_{t}G\left(\Gamma \right)\right)$. This can be rewritten in terms of the Dissipation Function [34](1)$$\Omega^{f_{t}}\left(\Gamma \right)\equiv -\left[\Lambda \left(\Gamma \right)+G\left(\Gamma \right)\cdot \nabla_{\Gamma }lnf_{t}\left(\Gamma \right)\right]$$
as (2)$$\frac{\partial f_{t}}{\partial t}\left(\Gamma \right)=f_{t}\Omega^{f_{t}}\left(\Gamma \right),$$
where $\Lambda =\nabla_{\Gamma }\cdot G$ is the phase space variation rate. The evolution of the ensemble average, defined by(3)$${\langle O\rangle }_{t}=\int_{M}O\left(\Gamma \right)\;f_{t}\left(\Gamma \right)\;d\Gamma$$
can be expressed with respect to the initial distribution $f_{0}$ as (4)$${\langle O\rangle }_{t}={\langle O\rangle }_{0}+\int_{0}^{t}{\langle \left(O\circ S^{s}\right)\Omega^{f_{0}}\rangle }_{0}\;ds.$$
See, e.g., Refs. [7,33,34,43] for detailed derivations. This result allows us to calculate the system’s response to an external perturbation in an exact, not approximate, way. Here, $f_{0}$ is the unperturbed distribution, which is usually the equilibrium ensemble for the unperturbed dynamics, while $S^{t}$ represents the exact (perturbed) dynamics. The origin of the name Dissipation Function comes from the fact that this quantity, for nonequilibrium molecular dynamics (NEMD), corresponds precisely to the energy dissipation rate: the product of dissipative force–time-associated flux. This is clear by comparing Expression (Equation 4) with those obtained using the TTCF for a thermodynamic system under the influence of an external field $F_{ext}$ [33]:(5)$${\langle O\rangle }_{t}={\langle O\rangle }_{0}-\frac{V}{k_{B}T}\int_{0}^{t}{\langle \left(O\circ S^{s}\right)J\cdot F_{ext}\rangle }_{0}\;ds.$$
Another interesting interpretation of Ω arises in the more general context of dynamical systems. Let $\Omega_{0,s}^{f_{0}}$ denote the time integral of the Dissipation Function between the time instants 0 and *s*:(6)$$\Omega_{0,s}^{f_{0}}\left(\Gamma \right)\equiv \int_{0}^{s}\Omega^{f_{0}}\left(S^{u}\Gamma \right)du$$
Then, one gets (7)$${\langle \Omega_{0,s}^{f_{0}}\rangle_{0}=\langle \ln\left(f_{0}/f_{-s}\right)\rangle }_{0}\geq 0$$
which is the relative entropy $D(f_{0}∥f_{-s})$ [34], or Kullback–Leibler divergence of the distributions $f_{0}$ and $f_{-s}$. This is of interest, for instance, in large deviation theory, where $\Omega^{f}$ plays the role of the large deviation functional, and has numerous consequences in applications [44]. Expression (Equation 4) also offers other advantages. Firstly, it keeps the probability fixed while allowing only the observables to evolve over time. Probability evolution requires the reversed dynamics, which is more difficult to use, whereas observables do not. For Equation (Equation 4), the dynamics are assumed to be invertible [34], although not necessarily time-reversal invariant, as often required in statistical mechanics. The use of the equilibrium distribution $f_{0}$ is fortunate because it is known and has the property of smoothing the result, effectively improving the signal-to-noise ratio [37]. In linear response theory, one uses $f_{0}$ as well, but in that case, the result is approximate rather than exact. Additionally, the notion of T-mixing, which is a necessary condition for the Fluctuation Relations to hold in nonequilibrium steady states [30], through its connection with the Dissipation Function (ΩT− mixing), (8)$${\langle \Omega^{f_{0}}\left(O\circ S^{t}\right)\rangle }_{0}\to {\langle \Omega^{f_{0}}\rangle }_{0}{\langle O\circ S^{t}\rangle }_{0}=0\;\mathrm{for}\;t\to \infty ,$$
provides a new approach for describing relaxation towards equilibrium [34].

The objective of this paper is to develop the quantum counterpart of the classical expression (Equation 4), since it could allow the extension to quantum mechanics of the benefits of the classical response theory. For sake of simplicity, we focus on finite-dimensional Hilbert spaces, allowing us to concentrate on constructing the new formalism while avoiding the technicalities of infinite dimensional analysis.

In quantum mechanics, a system is completely determined by its quantum state |ψ〉 which obeys the Schrödinger Equation [45]:(9)$$i\hbar \frac{\partial }{\partial_{t}}\psi \left(x,t\right)=H\psi \left(x,t\right)$$
a deterministic evolution of a probabilistic entity. In this case, one assumes that all degrees of freedom of interest are represented in the Hamiltonian H; hence, one refers to an isolated system. Observables are then expressed by self-adjoint operators [46]. The analogous notion of classical ensembles is here given by collections of states $|\psi_{j}\rangle$, each taken with probability $p_{j}$, with *j* running over a suitable set. Collectively, these states are represented by the density operator(10)$$\rho \equiv \sum_{j}p_{j}|\psi_{j}\rangle \langle \psi_{j}|$$
suitable for treating both pure and mixed states [46]. This operator is self-adjoint, $\rho^{†}=\rho$, has a unitary trace, Tr(ρ)=1, and is semi-definite positive, 〈ψ|ρ|ψ〉≥0∀ψ. The expectation value of an observable O is expressed through the trace operation, 〈O〉=Tr(ρO), which replaces the integral (Equation 3) of the classical theory.

There are two equivalent pictures [45]. In the Schrödinger picture, observables are time-independent, and ρ evolves over time according to the Von Neumann (Quantum Liouville) equation:(11)$$\frac{\partial }{\partial t}\;\rho_{t}=\frac{1}{i\hbar }\;\left[H,\rho_{t}\right].$$
Instead, in the Heisenberg picture, ρ remains fixed and the observables evolve in time according to(12)$$\frac{d}{dt}A_{t}=\frac{1}{i\hbar }\left[A_{t},H\right]+\partial_{t}A.$$
If the Hamiltonian is constant, the time propagator U(t) is defined as (13)$$U\left(t\right)\equiv e^{-\frac{it}{\hbar }H},\;\;\;U^{†}\left(t\right)=e^{+\frac{it}{\hbar }H}.$$
This time propagator is a one-parameter unitary group [46], i.e., it is unitary, $U\left(t\right)U^{†}\left(t\right)=U^{†}\left(t\right)U\left(t\right)=Id$, and it satisfies the group property U(t+s)=U(t)U(s) and U(0)=Id. Then, if the Hamiltonian is time-independent, the solutions of the previous equations can be written as (14)$$\left(\text{Schrödinger}\right)\;\rho_{t}=U\left(t\right)\rho_{0}U^{†}\left(t\right);\;\;\;(\mathrm{Heisenberg})\;A_{t}=U^{†}\left(t\right)AU\left(t\right).$$
These are the main tools we will use in the following.

## 3. Quantum Dissipation Function for Time-Independent Perturbations

We formulate the quantum exact response theory based on the Dissipation Function starting from scratch, so it is good to start by considering simple cases. Take a time-independent Hamiltonian $H_{0}$ and an arbitrary initial density operator $\rho_{0}$. Introduce a perturbation, producing a new Hamiltonian:(15)$$H=H_{0}+\theta \left(t\right)\lambda H_{ext},$$
where θ(t) is the Heaviside function. $H_{0}$ represents the equilibrium dynamics, and usually $\rho_{0}$ is the corresponding equilibrium density operator, which is stationary with respect to $H_{0}$:(16)$$[H_{0},\rho_{0}]=0.$$
The external perturbation $\lambda H_{ext}$ is assumed to be turned on at time t=0, and then kept constant over time, with λ∈R. Then, $H_{ext}$ makes $\rho_{0}$ no longer invariant, and the expectation values of the observables become time-dependent. Trying to translate the classical notion of Dissipation Function in this framework, we find there are some issues to tackle, such as the ordering problem. This requires us to choose a proper form for the Dissipation Function. From the classical expression (Equation 2), we obtain(17)$$\Omega^{f_{0}}\left(\Gamma \right)=f_{0}^{-1}\left(\Gamma \right)\frac{\partial f_{0}\left(\Gamma \right)}{\partial t}.$$
that holds only in the part of the phase space in which $f_{0}\left(\Gamma \right)>0$. This is reminiscent of the more general notion of absolute continuity of the evolved distributions with respect to the initial [47], and has been called ergodic consistency in molecular dynamics [43,48]. This limitation is not serious in many applications because the initial distribution usually corresponds to an unperturbed equilibrium state and does not vanish anywhere in the phase space of the perturbed evolution, if this satisfies the same constraints of the equilibrium dynamics [34,48].

The quantum mechanical counterpart of the classical Dissipation Function can then be guessed, using the von Neumann Equation (Equation 11), to be defined as follows:

**Definition** **1.**
*Let $\rho_{0}$ be an initial density operator and $H=H_{0}+\lambda H_{ext}$ a time-independent Hamiltonian operator. The Dissipation Operator can be defined as*

(18)
$$\Omega^{0}\equiv \frac{1}{i\hbar }\rho_{0}^{-1}\left[H,\rho_{0}\right].$$



As there are different equivalent definitions of the classical Dissipation Function, our choice is to some extent arbitrary, and other choices can be considered. We are going to show, however, that this form of $\Omega^{0}$ is consistent with linear response for small perturbations, and yields the exact solutions of problems whose exact solution is known. It remains that this definition faces two issues. First, the Dissipation Operator (Equation 18) is not self-adjoint, so its expectation value can be a complex number, ${\langle \Omega^{0}\rangle }_{t}\in C$. For isolated systems, this may be a challenge, since it does not represent a directly measurable quantity. However, this is not a problem per se in the case of open systems, where relying solely on self-adjoint operators may not be possible or necessary [49,50]. While it would be preferable to have a self-adjoint $\Omega^{0}$, this is not a priority. The second issue is that $\rho_{0}$ is not always invertible, as in the case for pure states. There are, however, ways to handle such a difficulty. One approach is to reduce the dimension of the Hilbert space, if one knows that the evolution remains within a subspace in which $\rho_{0}$ is invertible. Another way is to add arbitrarily small amounts $p_{\epsilon }$ to part of the entries of $\rho_{0}$ so that its rank becomes full, and later analyze the results in the $p_{\epsilon }\to 0$ limit. In any event, it is the same issue known in classical mechanics, and, analogously, the problem is solved if the initial distribution has support wider than that of the evolved distributions [47,48]. Despite these issues, Definition (Equation 18) has various advantages. It allows us to derive a quantum expression analogous to Equation (Equation 4). Moreover, it satisfies two important properties already present in the classical context of the Dissipation Function.

**Proposition** **1.**
*The expectation value of the Dissipation Operator calculated with respect to the density operator $\rho_{0}$ is always zero, and its initial time derivative is always positive:*

(19)
$${\langle \Omega^{0}\rangle }_{0}=0,\;\;\;\frac{d}{ds}{\langle \Omega^{0}\rangle }_{s}{|}_{s=0}\geq 0.$$



**Proof** **of** **Proposition** **1**.(20)$${\langle \Omega^{0}\rangle }_{0}=Tr\left(\rho_{0}\left(\frac{i}{\hbar }\rho_{0}^{-1}\left[H,\rho_{0}\right]\right)\right)=\frac{i}{\hbar }Tr\left(Id\cdot \left[H,\rho_{0}\right]\right)$$
Now, we can use first the linear property and then the cyclic property of the trace:(21)$$\begin{array}{c} \frac{i}{\hbar }Tr\left(H\rho_{0}-\rho_{0}H\right)=\frac{i}{\hbar }\left(Tr\left(H\rho_{0}\right)-Tr\left(\rho_{0}H\right)\right)=\frac{i}{\hbar }\left(Tr\left(H\rho_{0}\right)-Tr\left(H\rho_{0}\right)\right)=0 \end{array}$$
We conclude that ${\langle \Omega^{0}\rangle }_{0}=0$.We can show the second property by making the incremental limit explicit in the derivative operation:(22)$$\begin{array}{c} \frac{d}{ds}{\langle \Omega^{0}\rangle }_{s}{|}_{s=0}=\lim_{s\to 0}\frac{1}{s}\left[{\langle \Omega^{0}\rangle }_{s}-{\langle \Omega^{0}\rangle }_{0}\right]=\lim_{s\to 0}\frac{1}{s}\left[Tr\left(\rho_{s}\Omega^{0}\right)-Tr\left(\rho_{0}\Omega^{0}\right)\right]\\ =Tr\left(\lim_{s\to 0}\frac{\rho_{s}-\rho_{0}}{s}\Omega^{0}\right)=Tr\left(\frac{\partial \rho_{0}}{\partial t}\Omega^{0}\right) \end{array}$$
We now use a result that we will show later in Equation (Equation 43): we can also use the adjoint of the Dissipation Operator, ${\left(\Omega^{0}\right)}^{†}$, to express $\partial_{t}\rho_{0}$:(23)$$\frac{\partial \rho_{0}}{\partial t}={\left(\Omega^{0}\right)}^{†}\rho_{0}$$
Substituting this into the last expression, we obtain(24)$$\frac{d}{ds}{\langle \Omega^{0}\rangle }_{s}{|}_{s=0}=Tr\left(\frac{\partial \rho_{0}}{\partial t}\Omega^{0}\right)=Tr\left({\left(\Omega^{0}\right)}^{†}\rho_{0}\Omega^{0}\right)\geq 0$$
where the last inequality holds for the semi-positivity of $\rho_{0}$. In fact, let $A\in C^{n\times n}$ be a self-adjoint and semi-definite positive matrix and $C\in C^{n\times m}$ be an arbitrary matrix, then,(25)$$\mathrm{Tr}\left(C^{†}AC\right)=\sum_{i=1}^{m}{\left(C^{†}AC\right)}_{ii}=\sum_{i=1}^{m}c_{i}^{†}Ac_{i}\geq 0$$
where $c_{i}$ are the columns of *C*, and the last inequality follows from the definition of a semi-definite positive matrix.  □

We now derive one of the key results of this paper: an exact expression for the expectation value of observables, based on the new Dissipation Operator (Equation 18).

**Proposition** **2.**
*Let $\rho_{0}$ be the initial density operator and $H=H_{0}+\lambda H_{ext}$ the time-independent Hamiltonian operator. The expectation value of any observable O can be calculated exactly using the Dissipation Operator as*

(26)
$${\langle O\rangle }_{t}={\langle O\rangle }_{0}+\int_{0}^{t}{\langle \Omega^{0}O_{s}\rangle }_{0}ds$$



**Proof** **of** **Proposition** **2**.Let us derive a useful expression for $\frac{d}{ds}{\langle O\rangle }_{s}$:(27)$$\begin{array}{c} \frac{d}{ds}{\langle O\rangle }_{s}=\lim_{h\to 0}\frac{1}{h}\left[{\langle O\rangle }_{s+h}-{\langle O\rangle }_{s}\right]=\lim_{h\to 0}\frac{1}{h}\left[Tr\left(O\rho_{s+h}\right)-Tr\left(O\rho_{s}\right)\right]\\ =\lim_{h\to 0}\frac{1}{h}\left[Tr\left(OU\left(s+h\right)\rho_{0}U^{†}\left(s+h\right)\right)-Tr\left(OU\left(s\right)\rho_{0}U^{†}\left(s\right)\right)\right]\\ =\lim_{h\to 0}\frac{1}{h}\left[Tr\left(U^{†}\left(s+h\right)OU\left(s+h\right)\rho_{0}\right)-Tr\left(U^{†}\left(s\right)OU\left(s\right)\rho_{0}\right)\right]. \end{array}$$
Thanks to the group property of the operator U(t) for time-independent Hamiltonians, (28)$$U\left(s+h\right)=U\left(h\right)U\left(s\right),\;\;\;U^{†}\left(s+h\right)=U^{†}\left(s\right)U^{†}\left(h\right)\;,$$
thanks to the fact that observables and density matrix evolve “at opposite times”, and introducing the evolved operators $O_{s}=U^{†}\left(s\right)OU\left(s\right)$ we can write (29)$$\begin{array}{ccc} & & \;\lim_{h\to 0}\frac{1}{h}\left[Tr\left(U^{†}\left(h\right)O_{s}U\left(h\right)\rho_{0}\right)-Tr\left(O_{s}\rho_{0}\right)\right] \end{array}$$
(30)$$\begin{array}{ccc} & = & \lim_{h\to 0}\frac{1}{h}\left[Tr\left(O_{s}U\left(h\right)\rho_{0}U^{†}\left(h\right)\right)-Tr\left(O_{s}\rho_{0}\right)\right] \end{array}$$
(31)$$\begin{array}{ccc} & = & \lim_{h\to 0}\frac{1}{h}[Tr\left(O_{s}\left(U\left(h\right)\rho_{0}U^{†}\left(h\right))-\rho_{0}\right)\right] \end{array}$$
(32)$$\begin{array}{ccc} & = & Tr\left(O_{s}\lim_{h\to 0}\frac{\rho_{h}-\rho_{0}}{h}\right)=Tr\left(O_{s}\frac{\partial \rho_{0}}{\partial t}\right) \end{array}$$
From the definition (Equation 18) of the Dissipation Operator, we have(33)$$\frac{\partial \rho_{0}}{\partial t}=\rho_{0}\Omega^{0}$$
and substituting into (32), we obtain (34)$$Tr\left(O_{s}\frac{\partial \rho_{0}}{\partial t}\right)=Tr\left(O_{s}\rho_{0}\Omega^{0}\right)=Tr\left(\rho_{0}\Omega^{0}O_{s}\right)={\langle \Omega^{0}O_{s}\rangle }_{0}.$$
Then, we have (35)$$\frac{d}{ds}{\langle O\rangle }_{s}={\langle \Omega_{0}O_{s}\rangle }_{0}.$$
and (36)$${\langle O\rangle }_{t}={\langle O\rangle }_{0}+\int_{0}^{t}\frac{d}{ds}{\langle O\rangle }_{s}ds={\langle O\rangle }_{0}+\int_{0}^{t}{\langle \Omega^{0}O_{s}\rangle }_{0}ds$$  □

First, we note the strong similarity with the analogous Expression (Equation 4) in classical statistical mechanics. The system’s response is expressed in terms of the correlation function of the Dissipation Operator $\Omega^{0}$ and the observable $O_{t}$, evolved according to the exact dynamics H, calculated with respect to the equilibrium density operator $\rho_{0}$. This use of the initial distribution is common to the linear response, but like in the classical case, Equation (Equation 26) is not an approximate expression and is not limited to small perturbations.

In general, $\Omega^{0}$ and O do not commute, ${\langle \Omega^{0}O_{s}\rangle }_{0}\neq {\langle O_{s}\Omega^{0}\rangle }_{0}.$ Therefore, we cannot use ${\langle O_{s}\Omega^{0}\rangle }_{0}$ in Equation (Equation 26). In quantum mechanics, other types of correlations are often used, such as Kubo’s canonical correlation or symmetric correlation [3], in order to obtain real numbers as results, but for Expression (Equation 26), this is automatically obtained. Thus, the meaning of the quantum response function can be assigned to the form used in Equation (Equation 26). Nevertheless, we now consider a symmetrized version of our response formula, introducing the following notation:(37)$${\langle A;B\rangle }_{0}=Tr\left[\rho_{0}\;\frac{AB+BA}{2}\right].$$

## 4. Self-Adjoint Quantum Dissipation Operator

For the isolated time-independent dynamics considered above, the response of an observable ${\langle O\rangle }_{t}$ is computable in the Heisenberg and Schrödinger pictures without any need to introduce the Dissipation Function. In this sense, Expression (Equation 26) only represents a different formalism for expressing a well-known result. However, the different perspective it offers may be useful in the solution of complex time-independent problems, as it happens in classical mechanics, especially in the presence of poor signal-to-noise ratios [37]. Furthermore, its analogy with the classical Expression (Equation 4) may prove useful in time-dependent situations. Additionally, as in Equation (Equation 5), there may be opportunities to associate the Dissipation Operator $\Omega^{0}$ with the production of generalized entropy, involving, for instance the notion of Kullback–Leibler divergence (Equation 7). These interpretations are supported by the fact that Proposition 1 seems to imply the existence of h>0 such that (38)$${\langle \Omega_{0,s}^{0}\rangle }_{0}\geq 0\;\;\;\mathrm{for}\;\;0<s<h.$$
Finally, it could be possible to use the ΩT− mixing property [34,43] to study the relaxation of observables toward stationary states. This question would be especially interesting in the context of open quantum systems. Choosing O=1, we find an excellent consistency condition: the probabilistic interpretation of the density operator is preserved using Expression (Equation 26).

To explore this possibility, let us start observing that the minimum requirements of response theory are satisfied by the notions introduced above. In particular, Proposition 1 implies(39)$${\langle 1\rangle }_{t}={\langle 1\rangle }_{0}+\int_{0}^{t}{\langle \Omega^{0}1\rangle }_{0}ds=1+0t=1.$$
and, for a perturbation $H_{ext}$ that commutes with $\rho_{0}$, we have $\Omega^{0}=0$, hence (40)$${\langle O\rangle }_{t}={\langle O\rangle }_{0}+\int_{0}^{t}{\langle 0\rangle }_{0}ds={\langle O\rangle }_{0}$$
for all observables, as desired.

In the case where we take $O=\Omega^{0}$, and we wish to obtain a real number, as the dissipation should be, we have however a difficulty: $\Omega^{0}$ is not self-adjoint and ${\langle \Omega^{0}\rangle }_{t}$ may be complex. While this is not necessarily a problem for the theory of open systems, it is interesting to develop a self-adjoint Dissipation Operator. Therefore, we propose the following symmetrized operator:(41)$${\tilde{\Omega }}^{0}\equiv \frac{\Omega^{0}+{\left(\Omega^{0}\right)}^{†}}{2}.$$
as the Dissipation Function. Then, we note that(42)$$\begin{array}{ccc} {\left(\Omega^{0}\right)}^{†} & = & {\left(\frac{1}{i\hbar }\rho_{0}^{-1}\left[H,\rho_{0}\right]\right)}^{†}={\left(\frac{1}{i\hbar }\rho_{0}^{-1}H\rho_{0}-\frac{1}{i\hbar }H\right)}^{†}=-\frac{1}{i\hbar }{\left(\rho_{0}^{-1}H\rho_{0}\right)}^{†}+\frac{1}{i\hbar }H^{†}\\ & = & -\frac{1}{i\hbar }\left(\rho_{0}H\rho_{0}^{-1}-H\right)=-\frac{1}{i\hbar }\left[\rho_{0},H\right]\rho_{0}^{-1}=\frac{1}{i\hbar }\left[H,\rho_{0}\right]\rho_{0}^{-1} \end{array}$$
so that (43)$$\frac{\partial \rho_{0}}{\partial t}={\left(\Omega^{0}\right)}^{†}\rho_{0}$$
which leads to the following:

**Proposition** **3.***By means of the self-adjoint operator* (Equation 41)*, Equation* (Equation 26) *for time-independent perturbations can be expressed as*
(44)$${\langle O\rangle }_{t}={\langle O\rangle }_{0}+\int_{0}^{t}\left({\langle {\tilde{\Omega }}^{0};O_{s}\rangle }_{0}+\frac{1}{2}{\langle \left[\bar{\Omega^{0}},O_{s}\right]\rangle }_{0}\right)ds,$$
*where $\bar{\Omega^{0}}$ is the anti-self-adjoint operator $\bar{\Omega^{0}}\equiv \frac{1}{2}\left(\Omega^{0}-{\left(\Omega^{0}\right)}^{†}\right).$*

**Proof.** From the proof of (Equation 26), we know(45)$$\frac{d}{ds}{\langle O\rangle }_{s}=Tr\left(O_{s}\frac{\partial \rho_{0}}{\partial t}\right)$$
To introduce the Hermitian Dissipation Operator ${\tilde{\Omega }}^{0}$, we use (46)$$\frac{\partial \rho_{0}}{\partial t}=\rho_{0}\Omega^{0};\;\;\;\;\;\;\;\;\;\;\frac{\partial \rho_{0}}{\partial t}={\left(\Omega^{0}\right)}^{†}\rho_{0}.$$
Therefore, we can write (47)$$\begin{array}{ccc} Tr\left(O_{s}\frac{\partial \rho_{0}}{\partial t}\right) & = & \frac{1}{2}Tr\left(O_{s}\frac{\partial \rho_{0}}{\partial t}\right)+\frac{1}{2}Tr\left(O_{s}\frac{\partial \rho_{0}}{\partial t}\right)\\ & = & \frac{1}{2}Tr\left(O_{s}\rho_{0}\Omega^{0}\right)+\frac{1}{2}Tr\left(O_{s}{\left(\Omega^{0}\right)}^{†}\rho_{0}\right)=\frac{1}{2}Tr\left(\Omega^{0}O_{s}\rho_{0}+{\left(\Omega^{0}\right)}^{†}\rho_{0}O_{s}\right). \end{array}$$
We can rewrite this last expression as(48)$$\begin{array}{c} \;Tr\left(\Omega^{0}O_{s}\rho_{0}+{\left(\Omega^{0}\right)}^{†}\rho_{0}O_{s}\right)=Tr\left(\Omega^{0}O_{s}\rho_{0}+{\left(\Omega^{0}\right)}^{†}\left(O_{s}\rho_{0}+\left[\rho_{0},O_{s}\right]\right)\right)\\ \;=Tr\left(\left(\Omega^{0}+{\left(\Omega^{0}\right)}^{†}\right)O_{s}\rho_{0}\right)+Tr\left({\left(\Omega^{0}\right)}^{†}\left[\rho_{0},O_{s}\right]\right)=2Tr\left({\tilde{\Omega }}^{0}O_{s}\rho_{0}\right)+Tr\left({\left(\Omega^{0}\right)}^{†}\left[\rho_{0},O_{s}\right]\right). \end{array}$$
where the last equality follows from the definition (Equation 41). We would like to eliminate the presence of ${\left(\Omega^{0}\right)}^{†}$. For this purpose, we note that with similar procedures, but using the commutator property for the first term in the trace, we can equivalently write (49)$$Tr\left(\Omega^{0}O_{s}\rho_{0}+{\left(\Omega^{0}\right)}^{†}\rho_{0}O_{s}\right)=Tr\left(\Omega^{0}\rho_{0}O_{s}+\Omega^{0}\left[O_{s},\rho_{0}\right]+{\left(\Omega^{0}\right)}^{†}\rho_{0}O_{s}\right).$$
We find (50)$$Tr\left(\Omega^{0}O_{s}\rho_{0}+{\left(\Omega^{0}\right)}^{†}\rho_{0}O_{s}\right)=2Tr\left({\tilde{\Omega }}^{0}\rho_{0}O_{s}\right)+Tr\left(\Omega^{0}\left[O_{s},\rho_{0}\right]\right).$$
Now, we combine Equations (Equation 48) and (Equation 50). Starting from Equation (Equation 47), we have(51)$$Tr\left(O_{s}\frac{\partial \rho_{0}}{\partial t}\right)=\frac{1}{4}\left(Tr\left(\Omega^{0}O_{s}\rho_{0}+{\left(\Omega^{0}\right)}^{†}\rho_{0}O_{s}\right)+Tr\left(\Omega^{0}O_{s}\rho_{0}+{\left(\Omega^{0}\right)}^{†}\rho_{0}O_{s}\right)\right).$$
We can use Equation (Equation 48) in place of the first term and Equation (Equation 50) in place of the second one, obtaining(52)$$\begin{array}{c} \frac{1}{4}\left(2Tr\left({\tilde{\Omega }}^{0}O_{s}\rho_{0}\right)+Tr\left({\left(\Omega^{0}\right)}^{†}\left[\rho_{0},O_{s}\right]\right)+2Tr\left({\tilde{\Omega }}^{0}\rho_{0}O_{s}\right)+Tr\left(\Omega^{0}\left[O_{s},\rho_{0}\right]\right)\right)\\ =\frac{1}{2}Tr\left({\tilde{\Omega }}^{0}O_{s}\rho_{0}\right)+\frac{1}{2}Tr\left({\tilde{\Omega }}^{0}\rho_{0}O_{s}\right)+\frac{1}{4}\left(Tr\left({\left(\Omega^{0}\right)}^{†}\left[\rho_{0},O_{s}\right]+\Omega^{0}\left[O_{s},\rho_{0}\right]\right)\right)\\ =\frac{1}{2}Tr\left({\tilde{\Omega }}^{0}O_{s}\rho_{0}\right)+\frac{1}{2}Tr\left(O_{s}{\tilde{\Omega }}^{0}\rho_{0}\right)+\frac{1}{4}\left(Tr\left(\left(\Omega^{0}-{\left(\Omega^{0}\right)}^{†}\right)\left[O_{s},\rho_{0}\right]\right)\right). \end{array}$$
We can clearly use the symmetric correlation (Equation 37) for terms with ${\tilde{\Omega }}^{0}$, but we are unable to completely replace $\Omega^{0}$ and ${\left(\Omega^{0}\right)}^{†}$ with the hermitian ${\tilde{\Omega }}^{0}$. We can also observe that $\Omega^{0}-{\left(\Omega^{0}\right)}^{†}$ is the anti-self-adjoint part of $\Omega^{0}$:(53)$$\bar{\Omega^{0}}\equiv \frac{1}{2}\left(\Omega^{0}-{\left(\Omega^{0}\right)}^{†}\right).$$
In conclusion, we obtain (54)$$\frac{d}{ds}{\langle O\rangle }_{s}={\langle {\tilde{\Omega }}^{0};O_{s}\rangle }_{0}+\frac{1}{2}Tr\left({\bar{\Omega }}^{0}\left[O_{s},\rho_{0}\right]\right)={\langle {\tilde{\Omega }}^{0};O_{s}\rangle }_{0}+\frac{1}{2}{\langle \left[\bar{\Omega^{0}},O_{s}\right]\rangle }_{0}.$$
Integrating as already conducted previously, we obtain (55)$${\langle O\rangle }_{t}={\langle O\rangle }_{0}+\int_{0}^{t}\left({\langle {\tilde{\Omega }}^{0};O_{s}\rangle }_{0}+\frac{1}{2}{\langle \left[\bar{\Omega^{0}},O_{s}\right]\rangle }_{0}\right)ds.$$  □

This is an alternative form of Expression (Equation 26), which yields equivalent exact results. Since we are still in the early stages of formulating this theory, it is not clear which form best represents the method based on the Dissipation Operator. Therefore, we present both formulas. Equation (Equation 26) is in a simpler form, and its analogy with the classical expression is immediately apparent. On the other hand, the analogy between Expression (Equation 44) and the classical counterpart is less obvious, but it has the advantage of involving only self-adjoint (and anti-self-adjoint) operators. These operators are commonly used in quantum mechanics and possess well-known properties. Now, $\langle {\tilde{\Omega }}^{0}\rangle \in R$, and $\langle {\bar{\Omega }}^{0}\rangle$ is a pure imaginary number. Additionally, Expression (Equation 44) is based on the symmetric correlation (Equation 37), which shares characteristics much closer to the classical one. For instance, ${\langle {\tilde{\Omega }}^{0};O_{s}\rangle }_{0}={\langle O_{s};{\tilde{\Omega }}^{0}\rangle }_{0}$. Finally, with steps similar to those of the proof of Proposition 1, we obtain(56)$${\langle {\tilde{\Omega }}^{0}\rangle }_{0}={\langle \bar{\Omega^{0}}\rangle }_{0}=0;\;\;\;\;\;\;\frac{d}{ds}{\langle {\tilde{\Omega }}^{0}\rangle }_{s}{{|}_{s=0}\geq 0,\;\;\frac{d}{ds}{\langle \bar{\Omega^{0}}\rangle }_{s}|}_{s=0}=0,$$
which makes this new Dissipation Operator $\tilde{\Omega }$ also suitable as a candidate for being associated with the production of generalized entropy.

## 5. Applications to Qubits and Numerical Tests

In this section, we apply our expressions for the response to perturbations of qubit systems, and we compare with the results obtained using the Heisenberg picture. To perform these comparisons, we calculated the results numerically using MATLAB R2024a code. For mathematical convenience, we set ℏ=1 and treat all physical quantities as dimensionless.

A quantum system with only two energy states is referred to as a qubit. Qubits play a fundamental role in quantum computing, as they are the basic units of quantum information. They are described within two-dimensional Hilbert spaces. In this context, all possible physical observables are represented by linear combinations of the identity matrix and the Pauli matrices $\sigma_{i}$, while the density operator is expressed as the Bloch vector $\vec{a}$. One has (57)$$\begin{array}{c} H=\vec{c}\cdot {\vec{\sigma }}_{4},\;\;\;\;\;O=\vec{q}\cdot {\vec{\sigma }}_{4};\;\;\;\;\;\vec{c},\vec{q}\in R^{4}.\\ \rho =\frac{1}{2}\left(Id+\vec{a}\cdot \vec{\sigma }\right),\;\;\;\;\;\;\vec{a}\in R^{3},\;\left|\vec{a}\right|\leq 1. \end{array}$$
where $\vec{\sigma }=\left(\sigma_{x},\sigma_{y},\sigma_{z}\right)$ and ${\vec{\sigma }}_{4}=\left(Id,\vec{\sigma }\right)$, and(58)$$\sigma_{x}=\left[\begin{array}{cc} 0 & 1\\ 1 & 0 \end{array}\right],\;\sigma_{y}=\left[\begin{array}{cc} 0 & -i\\ i & 0 \end{array}\right],\;\sigma_{z}=\left[\begin{array}{cc} 1 & 0\\ 0 & -1 \end{array}\right],\;Id=\left[\begin{array}{cc} 1 & 0\\ 0 & 1 \end{array}\right].$$
We arbitrarily chose the vectors $\vec{a},\;\vec{c}$ and $\vec{q}$, and then we numerically simulated the subsequent evolution. This process was repeated for several different combinations of Hamiltonian, initial density, observable, and time interval. In all cases, Expressions (Equation 26) and (Equation 44) yielded equivalent results, coinciding with those of the Heisenberg and Schrödinger pictures, which are exact. Two examples of these tests are shown in Figure 1.

Because for some simple problems, approximate theories yield exact results [51], it is interesting to compare the new approach with linear response theory. Let us consider a spin-$\frac{1}{2}$ particle in a magnetic field $B_{z}$ directed along the *z*-axis. The dynamics are described by the Hamiltonian (59)$$H_{0}=\mu_{B}B_{z}\sigma_{z}=\frac{\hbar \omega_{0}}{2}\sigma_{z}.$$
The system is initially in equilibrium and described by the density operator (60)$$\rho_{0}=p_{↑}\left|↑\rangle \langle ↑\right|+p_{↓}\left|↓\rangle \langle ↓\right|=\left[\begin{array}{cc} p_{↑} & 0\\ 0 & p_{↓} \end{array}\right]$$
which is invariant under the unperturbed Hamiltonian, $[H_{0},\rho_{0}]=0$. The equilibrium dynamics is then disturbed by the interaction Hamiltonian $H_{ext}$:(61)$$H=H_{0}+\lambda H_{ext}=\frac{\omega_{0}}{2}\sigma_{z}+\lambda \left(\frac{\omega_{x}}{2}\sigma_{x}-\frac{\omega_{0}}{2}\sigma_{z}\right)=\left(1-\lambda \right)\frac{\omega_{0}}{2}\sigma_{z}+\lambda \frac{\omega_{x}}{2}\sigma_{x}$$
decreasing the z-direction component of the magnetic field, $B_{z}$, and producing a non-vanishing x-direction component $B_{x}$.

The linear response [3] considers perturbed dynamics of the form $H_{t}=H-K\left(t\right)A$, where K(t) is a time-dependent external force applied from the infinite past, t=−∞, when the system was at thermal equilibrium and described by an equilibrium density matrix $\rho_{e}$, i.e., $\rho \left(-\infty \right)=\rho_{e}$. *A* is a dynamical quantity conjugate to the applied force *K*. For this system, the linear response formula is (62)$$\begin{array}{c} {\langle B\rangle }_{t}={\langle B\rangle }_{\rho_{e}}+\Delta \bar{B}\left(t\right),\;\;\Delta \bar{B}\left(t\right)=\int_{-\infty }^{t}dt^{\prime }K\left(t^{\prime }\right)\phi_{BA}\left(t-t^{\prime }\right),\;\phi_{BA}\left(t\right)=\frac{1}{i\hbar }{\langle \left[A,B\left(t\right)\right]\rangle }_{\rho_{e}}; \end{array}$$
where *B* is an arbitrary observable and $B\left(t\right)=e^{\frac{i}{\hbar }Ht}Be^{-\frac{i}{\hbar }Ht}$ is the Heisenberg-evolved operator according to the unperturbed dynamics H (different from the total dynamics $H_{t}$). The dynamics in (Equation 61) are simpler than this, as we are considering a constant disturbance over time. We can assume the disturbance is absent before time t=0 and is impulsively turned on at t=0. Let us apply a change of notation from [3] to our notation:(63)$$A\to -H_{ext};\;K\left(t\right)\to \lambda \theta \left(t\right);\;\;\;\;H_{t}\to H=H_{0}+\lambda \theta \left(t\right)H_{ext};\;\;\;B\to O.$$
where θ(t) is the Heaviside function. Kubo’s formula can now be rewritten for our problem and simplified as(64)$$\begin{array}{c} {\langle \Delta O\rangle }_{t}=\int_{-\infty }^{t}dt^{\prime }\theta \left(t^{\prime }\right)\lambda \phi \left(t-t^{\prime }\right)=\int_{0}^{t}dt^{\prime }\lambda \phi \left(t-t^{\prime }\right)\\ =-\int_{t}^{0}dt^{*}\lambda \phi \left(t^{*}\right)=\int_{0}^{t}dt^{\prime }\lambda \phi \left(t^{\prime }\right). \end{array}$$
The response function is (65)$$\phi \left(t\right)=-\frac{1}{i\hbar }{\langle \left[H_{ext},e^{\frac{i}{\hbar }H_{0}t}Oe^{-\frac{i}{\hbar }H_{0}t}\right]\rangle }_{0}=-\frac{i}{\hbar }{\langle \left[U_{0}^{†}\left(t\right)OU_{0}\left(t\right),H_{ext}\right]\rangle }_{0},$$
where we used the subscript 0 in $U_{0}\left(t\right)$ to distinguish the evolution operator related to the equilibrium dynamics $H_{0}$ from U(t) of the total dynamic H: $U_{0}^{†}\left(t\right)OU_{0}\left(t\right)\neq O_{t}=U^{†}\left(t\right)OU\left(t\right)$. In conclusion, by setting ℏ=1, we obtain (66)$${\langle O\rangle }_{t}={\langle O\rangle }_{0}-i\lambda \int_{0}^{t}{\langle \left[U_{0}^{†}\left(t\right)OU_{0}\left(t\right),H_{ext}\right]\rangle }_{0}dt^{\prime }.$$
In Figure 2, we compare the results with different values of λ, $\rho_{0}$, time interval and observable O, for the expressions of $O=\sigma_{y}$ given by the linear theory, by the Heisenberg picture and by the Dissipation Function formalism. Qualitatively very similar results (as λ varies) were found for other self-adjoint observables and different initial density operators $\rho_{0}$ (with $\left[H_{0},\rho_{0}]=0\right)$.

For small λ, the linear approximation is accurate, and the difference between linear and $\Omega^{0}$ formulas is indistinguishable to the eye. As λ increases, the linear approximation becomes increasingly worse, as expected, while the $\Omega^{0}-$response continues to provide results consistent with those of the Heisenberg picture. Moreover, the linear response theory becomes less accurate as time increases, whereas the expression in (Equation 26) does not exhibit this flaw. This illustrates the limitations of linear theories even in simple systems. However, for small perturbations, the linear response can handle much more complex dynamics than these ones. It remains to be seen how useful the Dissipation Function can be in such situations.

## 6. Properties of Dissipation and Time-Dependent Perturbations

It is clear that the Dissipation Operator plays a very precise role in the temporal evolution of the density operator $\rho_{t}$: (67)$$\frac{\partial \rho_{0}}{\partial t}=\rho_{0}\Omega^{0}.$$
However, we can consider any time $t^{*}$ as the initial instant of the dynamics and, consequently, define (68)$$\Omega^{t^{*}}=\frac{1}{i\hbar }\rho_{t^{*}}^{-1}\left[H,\rho_{t^{*}}\right]$$
which allows us to write the Von Neumann Equation (Equation 11) as (69)$$\begin{array}{c} \frac{\partial \rho_{t}}{\partial t}=\rho_{t}\Omega^{t};\;\;\;\;\;\;\rho \left(0\right)=\rho_{0}. \end{array}$$
In general, the formal solution of an equation of this form is expressed using the anti-time-ordering operator $T_{R}$ [52]: (70)$$\rho_{t}=\rho_{0}T_{R}\left[e^{\int_{0}^{t}\Omega^{s}ds}\right].$$
To make use of Equation (Equation 70), we need to express $\Omega^{s}$ without explicitly relying on $\rho_{s}$, which is contained in the definition of the Dissipation Operator. This can be carried out for constant Hamiltonian dynamics.

**Proposition** **4.**
*If the Hamiltonian H is time-independent, $\Omega^{s}$ coincides with the initial Dissipation Operator $\Omega^{0}$ evolved backward in time:*

(71)
$$\Omega^{s}=U\left(s\right)\Omega^{0}U^{†}\left(s\right).$$



**Proof.** By definition, we have (72)$$\Omega^{s}=\frac{1}{i\hbar }\rho_{s}^{-1}\left[H,\rho_{s}\right].$$
We can express $\rho_{s}$ using time propagators:(73)$$\begin{array}{c} \Omega^{s}=\frac{1}{i\hbar }{\left(U\left(s\right)\rho_{0}U^{†}\left(s\right)\right)}^{-1}\left[H,U\left(s\right)\rho_{0}U^{†}\left(s\right)\right]\\ =\frac{1}{i\hbar }{\left(U^{†}\left(s\right)\right)}^{-1}\rho_{0}^{-1}U^{-1}\left(s\right)\left[H,U\left(s\right)\rho_{0}U^{†}\left(s\right)\right]\\ =\frac{1}{i\hbar }U\left(s\right)\rho_{0}^{-1}U^{†}\left(s\right)\left(HU\left(s\right)\rho_{0}U^{†}\left(s\right)-U\left(s\right)\rho_{0}U^{†}\left(s\right)H\right)\\ =\frac{1}{i\hbar }U\left(s\right)\rho_{0}^{-1}U^{†}\left(s\right)HU\left(s\right)\rho_{0}U^{†}\left(s\right)-\frac{1}{i\hbar }U\left(s\right)\rho_{0}^{-1}U^{†}\left(s\right)U\left(s\right)\rho_{0}U^{†}\left(s\right)H; \end{array}$$
and this expression can be simplified noting that(74)$$U\left(s\right)\rho_{0}^{-1}U^{†}\left(s\right)U\left(s\right)\rho_{0}U^{†}\left(s\right)=Id,\;\;\mathrm{and}\;\;\left[U\left(t\right),H\right]=0.$$
We can then write (75)$$\begin{array}{c} \Omega^{s}=\frac{1}{i\hbar }U\left(s\right)\rho_{0}^{-1}H\rho_{0}U^{†}\left(s\right)-\frac{1}{i\hbar }U\left(s\right)U^{†}\left(s\right)HU\left(s\right)U^{†}\left(s\right)\\ =\frac{1}{i\hbar }U\left(s\right)\left(\rho_{0}^{-1}H\rho_{0}-U^{†}\left(s\right)HU\left(s\right)\right)U^{†}\left(s\right)\\ =\frac{1}{i\hbar }U\left(s\right)\left(\rho_{0}^{-1}H\rho_{0}-H\right)U^{†}\left(s\right)\\ =U\left(s\right)\left(\frac{1}{i\hbar }\rho_{0}^{-1}\left[H,\rho_{0}\right]\right)U^{†}\left(s\right)=U\left(s\right)\Omega^{0}U^{†}\left(s\right). \end{array}$$  □

Interestingly, the Dissipation Operator does not evolve in time like standard observables in the Heisenberg representation, $U^{†}\left(t\right)OU\left(t\right)$. This might seem problematic: we want to consider Ω as a physical observable, but it evolves like a probability, i.e., in reverse time: $\Omega^{t}=U\left(t\right)\Omega^{0}U^{†}\left(t\right)$. This is not due to the order of its operators in the definition. One could try to redefine the Dissipation Operator by swapping the order of $H,\;\rho_{0}$ and $\rho_{0}^{-1}$, but it would still evolve in reversed time. The origin of this type of evolution lies in the operators involved: H and ρ. The Hamiltonian operator H (in this simple case) commutes with time propagators, while ρ and $\rho^{-1}$ evolve at reversed times.

If we want a Dissipation Operator that evolves like any other physical observable, we should avoid the use of ρ. At the moment, this is not of our concern. Indeed, upon further analysis, this result does not seem to be a disadvantage at all; on the contrary, it looks more a strength. It provides consistent results, it is equivalent with the Heisenberg and Schrödinger pictures for usual observables. Using $\Omega^{0}$, we found (76)$$\frac{d}{ds}{\langle O\rangle }_{s}=Tr\left(\rho_{0}\Omega^{0}O_{s}\right)=Tr\left(O\frac{\partial \rho_{s}}{\partial s}\right)$$
which conceptually is reminiscent of the Heisenberg picture, with evolving observables, and of the Schrödinger picture with evolving probabilities. In turn, the equivalence of the two is obtained thanks to the reverse-time evolution of $\Omega^{0}$:(77)$$\begin{array}{c} {\left(\frac{d}{ds}{\langle O\rangle }_{s}\right)}^{S}=Tr\left(O\frac{\partial \rho_{s}}{\partial s}\right)=Tr\left(O\rho_{s}\Omega^{s}\right)=Tr\left(OU\left(s\right)\rho_{0}U^{†}\left(s\right)U\left(s\right)\Omega^{0}U^{†}\left(s\right)\right)\\ =Tr\left(U^{†}\left(s\right)OU\left(s\right)\rho_{0}\Omega^{0}\right)=Tr\left(O_{s}\rho_{0}\Omega^{0}\right)=Tr\left(\rho_{0}\Omega^{0}O_{s}\right)={\left(\frac{d}{ds}{\langle O\rangle }_{s}\right)}^{H}. \end{array}$$
where the superscript *S* in the first term and the superscript *H* in the last one respectively stand for Schrödinger and Heisenberg. Moreover, Equation (Equation 71) yields another consistency result:(78)$${\langle \Omega^{t}\rangle }_{t}=Tr\left(U\left(t\right)\Omega^{0}U^{†}\left(t\right)U\left(t\right)\rho_{0}U^{†}\left(t\right)\right)={\langle \Omega^{0}\rangle }_{0}=0$$
which correctly means that every moment can be considered as the initial one. The property of zero-mean, stated in Proposition 1, is propagated in time thanks to Equation (Equation 71). Finally, in applications of classical dynamical systems, a commonly used quantity is ${\langle \Omega^{0}\rangle }_{t}$ [35], but here, ${\langle \Omega^{0}\rangle }_{t}\neq {\langle \Omega^{t}\rangle }_{0}$, unlike what happens to the usual observables.

Equation (Equation 70) shows that the solution to the Von Neumann problem (Equation 11) can be expressed in several ways: not only in the usual form, Equation (Equation 14), but also as in Equation (Equation 70), or equivalently, using the adjoint operator $\Omega^{†}$ and the time-ordering operator $T_{L}$, as (79)$$\rho_{t}=T_{L}\left[e^{\int_{0}^{t}{\left(\Omega^{†}\right)}^{s}ds}\right]\rho_{0}.$$
If the solution is unique, Expressions (Equation 70) and (Equation 79) coincide with the usual form (Equation 14); they are just different ways of writing the same thing.

Applying our theory to complex, particularly open quantum systems for which traditional time propagators do not exist, could prove useful. Let us investigate this possibility. Take a time-dependent Hamiltonian H(t). The usual time evolution is expressed by [52](80)U†(t)=TRexp+iℏ∫0tH(s)ds;U(t)=TLexp−iℏ∫0tH(s)ds;ρt=U(t)ρ0U†(t).aaaaaaaaaaaaaaaaaaaaaaaaabbbbbbbbbbbbbbbbbbbbbbbbb
In terms of Dissipation Operators, one can instead follow different approaches. We choose here self-adjoint Dissipation Operators, introducing the operator $\omega^{0}\equiv \hbar^{-1}\rho_{0}^{-1}H\rho_{0}$, and then we symmetrize it to make it self-adjoint:(81)$${\tilde{\omega }}^{0}=\frac{1}{2\hbar }\left(\rho_{0}^{-1}H\rho_{0}+\rho_{0}H\rho_{0}^{-1}\right).$$
We have: (82)$$\begin{array}{c} \bar{\Omega^{0}}=\frac{1}{2}\left(\Omega^{0}-{\left(\Omega^{0}\right)}^{†}\right)=\frac{1}{2i\hbar }\left(\rho_{0}^{-1}\left[H,\rho_{0}\right]-\left[H,\rho_{0}\right]\rho_{0}^{-1}\right)\\ =\frac{1}{2i\hbar }\left(\rho_{0}^{-1}H\rho_{0}-2H+\rho_{0}H\rho_{0}^{-1}\right)=\frac{1}{i}\left({\tilde{\omega }}^{0}-\frac{1}{\hbar }H\right). \end{array}$$
Now, we can rewrite the Hamiltonian as $H\left(s\right)=\hbar \left({\tilde{\omega }}^{s}-i\bar{\Omega^{s}}\right)$ and define the operator ${\hat{\Omega }}^{s}\equiv {\tilde{\omega }}^{s}-i\bar{\Omega^{s}}=H/\hbar$. This allows us to write(83)$$\hat{U}\left(t\right)\equiv T_{L}e^{-i{\hat{\Omega }}_{0,t}},\;\;{\hat{U}}^{†}\left(t\right)=T_{R}e^{+i{\hat{\Omega }}_{0,t}}={\hat{U}}^{-1}\left(t\right);\;\;O_{t}={\hat{U}}^{†}\left(t\right)O\hat{U}\left(t\right).$$
Clearly, for constant Hamiltonians, these ordered exponentials become mere exponentials, because the two time-dependent operators $\bar{\Omega^{t}}$ and ${\tilde{\omega }}^{t}$ reduce to the constant operator H. While this is simply a rewriting of the usual operators, it suggests an interesting possibility: it may be possible to construct time propagators for dissipative dynamics by extending the Dissipation Operator formalism to open quantum systems.

## 7. The Dissipation Operator for Open Quantum Systems

The Lindblad equation [9](84)$$\dot{\rho }=\frac{1}{i\hbar }\left[H,\rho \right]+\sum_{\alpha }\left(L_{\alpha }\rho L_{\alpha }^{†}-\frac{1}{2}\left\{L_{\alpha }^{†}L_{\alpha },\rho \right\}\right),$$
with the commutator accounting for the unitary non-dissipative evolution and the $L_{\alpha }$ operators for the rest, is commonly used to describe open quantum systems. In particular, H is the effective Hamiltonian operator, which may differ from the Hamiltonian of the isolated system due to the interaction with the environment. The second term represents the dissipative part of the evolution, with the Lindblad operators $L_{\alpha }$ describing the interaction between the system and external environment. To apply our method to these dynamics, we extend the Dissipation Operator incorporating the dissipative effects introduced by the Lindblad operators. Again, there are several equivalent ways to achieve this; we adopt the following:(85)$$D_{\alpha }\equiv \rho^{-1}L_{\alpha }\rho L_{\alpha }^{†}-L_{\alpha }^{†}L_{\alpha }\Rightarrow \Omega_{L}^{0}\equiv \Omega^{0}+\sum_{\alpha }D_{\alpha }^{0}.$$
Here, $\Omega^{0}$ concerns as above the Hamiltonian part of the evolution, and $D_{\alpha }$ takes into account the dissipative part:(86)$$D_{\alpha }^{†}={\left(\rho^{-1}L_{\alpha }\rho L_{\alpha }^{†}\right)}^{†}-{\left(L_{\alpha }^{†}L_{\alpha }\right)}^{†}={\left(\rho L_{\alpha }^{†}\right)}^{†}{\left(\rho^{-1}L_{\alpha }\right)}^{†}-{\left(L_{\alpha }\right)}^{†}{\left(L_{\alpha }^{†}\right)}^{†}=L_{\alpha }\rho L_{\alpha }^{†}\rho^{-1}-L_{\alpha }^{†}L_{\alpha }$$
and linearity implies ${\left(\sum_{\alpha }D_{\alpha }\right)}^{†}=\sum_{\alpha }D_{\alpha }^{†}$. Then, the Lindblad equation can be rewritten as(87)$$\frac{\partial \rho }{\partial t}=\frac{1}{2}\left(\rho \Omega_{L}^{t}+{\left(\Omega_{L}^{t}\right)}^{†}\rho \right)$$
Indeed, we can write (88)$$\begin{array}{ccc} \frac{\partial \rho }{\partial t} & = & \frac{1}{2}\left(\rho \Omega_{L}^{t}+{\left(\Omega_{L}^{t}\right)}^{†}\rho \right)\\ & = & \frac{1}{2}\left(\rho \Omega^{t}+{\left(\Omega^{t}\right)}^{†}\rho \right)+\frac{1}{2}\sum_{\alpha }\left(L_{\alpha }\rho L_{\alpha }^{†}-\rho L_{\alpha }^{†}L_{\alpha }+L_{\alpha }\rho L_{\alpha }^{†}-L_{\alpha }^{†}L_{\alpha }\rho \right)\\ & = & \frac{1}{i\hbar }\left[H,\rho \right]+\sum_{\alpha }\left(L_{\alpha }\rho L_{\alpha }^{†}-\frac{1}{2}\left\{L_{\alpha }^{†}L_{\alpha },\rho \right\}\right). \end{array}$$
We note that, for Hamiltonian dynamics, one can express the Von Neumann equation with a linear combination of Ω and $\Omega^{†}$: (89)$$\frac{\partial \rho }{\partial t}=a{\left(\Omega \right)}^{†}\rho +b\rho \Omega \;\;\;\mathrm{with}\;a,b\in R\;s.t.\;a+b=1.$$
Applying definition (Equation 85) to the Lindblad equation, we take a=b=1/2, consistently with Equation (Equation 87). Various practical challenges now arise in explicitly solving the equations of interest. Therefore, to illustrate how our approach works, here, we introduce some simplifying hypotheses. In particular, we take (90)$$O_{t}=U_{L}^{\times }\left(t\right)OU_{L}\left(t\right)=e^{\frac{i}{\hbar }\left(H+C^{\times }\right)t}Oe^{-\frac{i}{\hbar }\left(H+C\right)t}.$$
where the symbol ‘×’ indicates that the two evolution operators need not necessarily be adjoints of each other. This hypothesis is suggested by the extension of $\hat{\Omega }$ in the form (91)$${\hat{\Omega }}_{L}^{s}=\left({\tilde{\omega }}^{s}-i\bar{\Omega^{s}}\right)+\left(g^{s}-k^{s}\right)=\frac{1}{\hbar }\left(H+C\right),$$
where $g(L_{\alpha },L_{\alpha }^{†},\rho_{s})$ and $k(L_{\alpha },L_{\alpha }^{†},\rho_{s})$ represent the dissipative part of ${\tilde{\omega }}_{L}$ and ${\bar{\Omega }}_{L}$, respectively. These are time-dependent operators. The term $k^{s}$ is determined by the definition of ${\bar{\Omega }}_{L}$, but the term $g^{s}$ has no specific constraints. This allows us to define $g^{s}$ consistently with the correct results. It is reasonable to express these terms through a general operator *C*, precisely because of this freedom with respect to *g*. Additionally, assuming that *C* is time-independent is not unrealistic. Just as the two time-dependent operators $\bar{\Omega^{t}}$ and ${\tilde{\omega }}^{t}$ reduce to the constant Hamiltonian H, the same could happen for $g^{t}$ and $k^{t}$, especially since all the Lindblad operators in Equation (Equation 84) are time-independent. We stress that in Equation (Equation 90), we are not assuming $U_{L}$ to be a “real” time propagator, in the sense that it would extend the Heisenberg representation, ${\langle O\rangle }_{t}=Tr\left(U_{L}^{\times }\left(t\right)OU_{L}\left(t\right)\rho_{0}\right)$. Instead, we are stating that these operators behave like evolution operators when combined with the generalization of expression (Equation 26). This generalization can be derived exactly by repeating the calculations already performed earlier, starting from $d_{t}{\langle O\rangle }_{t}=Tr\left(O_{t}^{*}\partial_{t}\rho_{0}\right)$, where the only difference is represented by Equation (Equation 87). This leads to(92)$${\langle O\rangle }_{t}={\langle O\rangle }_{0}+\frac{1}{2}\int_{0}^{t}\left({\langle \Omega_{L}^{0}O_{s}^{*}\rangle }_{0}+{\langle O_{s}^{*}{\left(\Omega_{L}^{0}\right)}^{†}\rangle }_{0}\right)ds.$$
Let us apply Expression (Equation 92) to a particular Lindblad equation, whose analytical solution is known, suitably defining *C* and $C^{\times }$, and consequently $U_{L}\left(t\right)$ and $U_{L}^{\times }\left(t\right)$. Consider the equation(93)$$\frac{\partial \rho }{\partial t}=-i\left[\omega \sigma_{z},\rho \right]+\gamma \left(\sigma^{-}\rho \sigma^{+}-\frac{1}{2}\left\{\sigma^{+}\sigma^{-},\rho \right\}\right);\;\;\sigma^{\pm }\equiv \frac{1}{2}\left(\sigma_{x}\pm i\sigma_{y}\right),$$
which describes a decoherence phenomenon of a qubit open system. This equation is already in the form of (Equation 84); we just need to highlight the correspondence with the general notation (94)$$\alpha =1;\;H=\omega \sigma_{z};\;\;L=\sqrt{\gamma }\sigma^{-};\;\;L^{†}=\sqrt{\gamma }\sigma^{+}.$$
After writing Equation (Equation 93) in its single components, the analytical solution can be easily calculated:(95)$$\rho \left(t\right)=\left[\begin{array}{cc} \rho_{00}^{0}e^{-\gamma t} & \rho_{01}^{0}e^{-(0.5\gamma +2i\omega )t}\\ \rho_{10}^{0}e^{-(0.5\gamma -2i\omega )t} & \rho_{11}^{0}+\rho_{00}^{0}\left(1-e^{-\gamma t}\right) \end{array}\right]$$
and then the evolution of observables, ${\langle O\rangle }_{t}=Tr\left(\rho \left(t\right)O\right)$ immediately follows. Comparing this expression with Equation (Equation 92), we find the form of *C* and $C^{\times }$ for this particular case. Consider first diagonal observables:(96)$$O^{1}=\left[\begin{array}{cc} a & 0\\ 0 & b \end{array}\right],\;\;a,b\in R.$$
Introducing (97)$$C\equiv -i\gamma \sigma^{+}\sigma^{-},\;C^{\times }\equiv +i\gamma \sigma^{-}\sigma^{+},$$
Formula (Equation 92) gives the known analytical solution (Equation 95), for any initial condition ρ(0) and for all real frequencies ω,γ. In Figure 3, this is shown numerically for the projectors $\pi_{0}$ (a=1,b=0) and $\pi_{1}$ (a=0,b=1), from which all diagonal observables can be obtained. It is important to note that $C^{\times }$ differs from the adjoint of *C*, $C^{†}=i\gamma \sigma^{+}\sigma^{-}$, and that the two operators $U_{L}$ and $U_{L}^{\times }$ do not form a unitary group. This is consistent with the dissipative nature of the dynamics.

Consider now the following class of observables:(98)$$O^{2}=\left[\begin{array}{cc} 0 & c+id\\ c-id & 0 \end{array}\right];\;c,d\in R.$$
which we call diag2 operators. We define the evolution of the observables for this second class as(99)$$O_{t}^{*}=e^{\frac{i}{\hbar }\left(H+M^{\times }\right)t}O^{2}e^{-\frac{i}{\hbar }\left(H+M\right)t};\;\;\;M=-\frac{i}{4}\left\{\sigma^{-},\sigma^{+}\right\}=-M^{\times }.$$
When combined with Expression (Equation 92), this provides the exact response for all observables of the form (Equation 98), with *M*, $M^{†}=M^{\times }$. In Figure 4, we numerically illustrate this fact for two cases.

Any self-adjoint observable O in a two-dimensional Hilbert space can be expressed as a linear combination of the two kinds of operators defined by Equations (Equation 96) and (Equation 98). For the specific case of Equation (Equation 93), the operators $C,\;C^{\times },\;M\;\mathrm{and}\;M^{\times }$ all commute with the Hamiltonian operator H. In conclusion, for this particular Lindblad Equation (Equation 93), the expression (Equation 92) gives exact results when the observables evolve according to (100)$$O_{t}^{*}=e^{\frac{i}{\hbar }C^{\times }t}O_{H}^{1}\left(t\right)e^{-\frac{i}{\hbar }Ct}+e^{\frac{i}{\hbar }M^{\times }t}O_{H}^{2}\left(t\right)e^{-\frac{i}{\hbar }Mt},$$
where a simple rule applies:(101)$$M=-\frac{i}{4}\left\{L_{\alpha },L_{\alpha }^{†}\right\}=-M^{\times };\;\;C=-iL_{\alpha }^{†}L_{\alpha };\;\;C^{\times }=iL_{\alpha }L_{\alpha }^{†}$$
with the subscript ‘*H*’ indicating evolution in the Heisenberg picture.

How generally the rule (Equation 101) applies is the object of our investigation. However, this is just one way to obtain exact results using the Dissipation Operator. Exact results can be obtained equivalently by evolving the observables in the Heisenberg picture and assuming the following time dependence for the Dissipation Operator:(102)$$\Omega_{L}^{0}\left(t\right)=e^{-\gamma t}\Omega_{L}^{0}\;\;\;\mathrm{for}\;\mathrm{diagonal}\;\mathrm{operators}\;;\;\Omega_{L}^{0}\left(t\right)=e^{-\frac{1}{2}\gamma t}\Omega_{L}^{0}\;\;\;\mathrm{for}\;\mathrm{diag}\;2\;\mathrm{operators}$$

Lindblad equations, while widely used, have certain limitations. One issue is that the positivity of the density matrix ρ can be violated, particularly in numerical simulations (1.5.2 in [9]). Moreover, their validity is restricted to linearly dissipative interactions; thus, choosing $L_{\alpha }$ operators can be an ambiguous or approximate process [53,54]. Furthermore, when many energy states are involved, calculating the solution becomes very computationally intensive [55,56]. We conclude by observing that Lindblad equations are intimately linked to the concept of quantum entropy [57,58], which is affected by dissipation, of course. Therefore, combining them with the response theory based on the Dissipation Operator may prove useful.

## 8. Concluding Remarks

In this paper, we have presented a first approach to the formulation of a quantum exact response theory as an extension of the classical theory based on the Dissipation Function. Our future goal is to further develop this approach for studying perturbations in complex open quantum systems. To this end, we have started from constant Hamiltonian dynamics, which suggest at least two quite natural quantum versions of the Dissipation Function, which are equivalent for the cases considered here.

The first quantum Dissipation Operator has been defined by (Equation 18), and the corresponding quantum exact response expression has been given by (Equation 26), in perfect analogy with the classical Formula (Equation 4). For constant Hamiltonian dynamics, the correctness of the results can be tested, comparing the expectation values of observables ${\langle O\rangle }_{t}$ with the corresponding Schrödinger and Heisenberg pictures.

We then noted that, although it is not strictly required for open dynamics, it is still interesting to use self-adjoint operators. Therefore, we proposed a self-adjoint Dissipation Operator, defined by Equation (Equation 44), which is equivalent to the first. In our derivation, this requires the introduction of an anti-self-adjoint operator. In this formulation, symmetric quantum correlations appear, with properties quite similar to the classical case.

We applied these expressions to qubit systems and obtained numerically correct results. A comparison with linear response theory revealed that as the perturbation value and time increase, the linear response naturally worsens, whereas the expressions (Equation 26) and (Equation 44) continue to yield exact results. We also examined the role of the Dissipation Operator in the time evolution of the density matrix ρ. In particular, we found that for constant Hamiltonian dynamics, the operator $\Omega^{t}$ evolves in reverse time, as described in Equation (Equation 71). This is not what observables do, but it is consistent with the classical theory, where the Dissipation Function is at once an observable and the generator of the evolution of probabilities. As far as response is concerned, this poses no problem.

As it is preferable to keep the Dissipation Operator $\Omega^{0}$ fixed in time, as in the classical theory, we introduced time propagators based on the Dissipation Operators, as given in Equation (Equation 83). We carried this out consistently, rewriting the Hamiltonian in terms of Dissipation Operators, which may seem trivial but suggests an interesting insight: by extending this approach, it may be possible to properly treat observables of dissipative dynamics. Building on this idea, we extended the method to Lindblad equations, incorporating the dissipative part (described by $L_{\alpha }$ and $L_{\alpha }^{†}$) in the definition of the Dissipation Operator, leading to the quantum exact response Expression (Equation 92). We applied this approach to a specific Lindblad equation, comparing the results obtained through our new expression with the known analytical solution. With the aid of assumptions suggested by both previous studies and by inspection, we obtained exact results with Expression (Equation 92).

While this approach is still in its infancy, and it is open to different approaches, it offers an original perspective on quantum response theory and open quantum systems. Its main strength lies in its analogy to the classical exact response theory, which is robust and general [31,33,35,36,37]. Moreover, the computational advantages recently offered by classical theory across various applications [37,59], along with the development of new methods for investigating the Dissipation Function [41], provide further compelling evidence of the potential held by its proposed quantum analogue. A promising avenue for future research is to investigate whether Ω can be associated with entropy production rates, and with the mathematical tools of large deviation theory, such as the Kullback–Leibler divergence. Future studies intend to clarify this, also in connection with the application to different equations for open quantum systems. Finally, extension to infinite-dimensional Hilbert spaces should be considered.

## Figures and Tables

**Figure 1 entropy-27-00527-f001:**
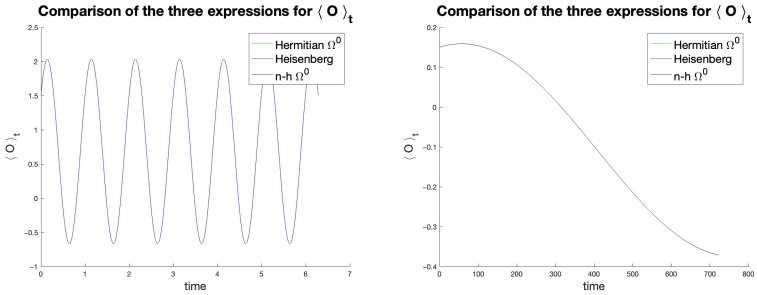
(**Left**): $\vec{a}=\left(0.3,0.3,-0.3\right)$; $\vec{c}=\left(4,0.8,0.5,3\right)$; $\vec{q}=\left(1,0.2,3,1.5\right)$. (**Right**): $\vec{a}=\left(0,0.1,-0.5\right)$; $\vec{c}=\left(10^{-3},10^{-3},2\cdot 10^{-3},0\right)$; $\vec{q}=\left(-0.1,-0.2,0,-0.5\right)$. Green: ${\tilde{\Omega }}^{0}$ expression; blue: $\Omega^{0}$ expression; red: Heisenberg expression. The three curves always overlap.

**Figure 2 entropy-27-00527-f002:**
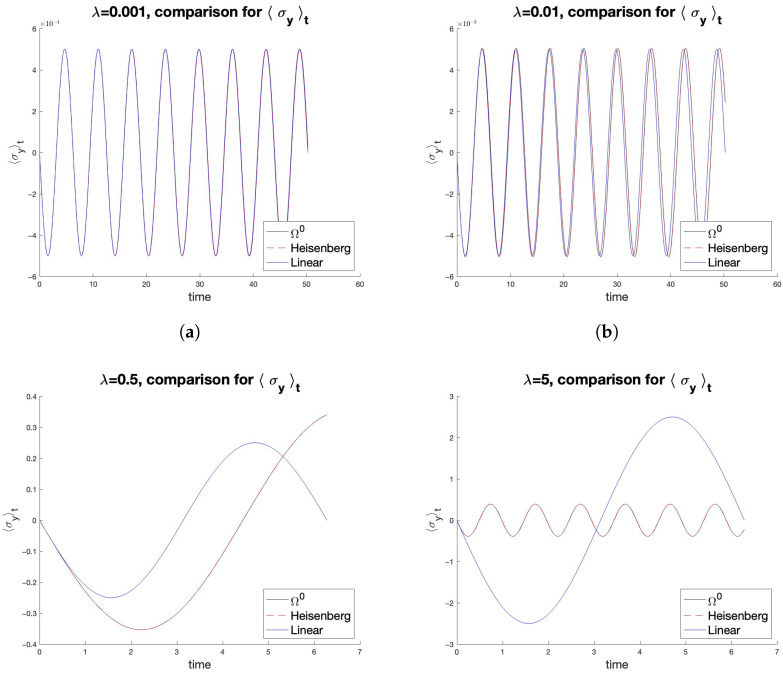
Comparison between linear (blue), Heisenberg (red) and dissipation (black) responses for increasing perturbation values. In all panels, we have $O=\sigma_{y}$, $p_{↑}=0.75$, $p_{↓}=0.25$, $\omega_{x}=\omega_{0}=1$. Ω− Expression (Equation 26) always coincides with the Heisenberg picture (the two curves overlap); the linear response does not. (**a**) λ=0.001, the three responses coincide. (**b**) λ=0.01, the linear response differs slightly over long times. (**c**) λ=0.5 and (**d**) λ=5, the difference is big.

**Figure 3 entropy-27-00527-f003:**
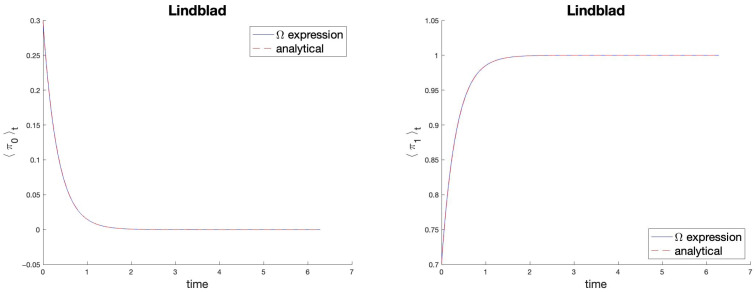
Comparison of the results offered by Expression (Equation 92) (solid blue line) with those known theoretically (dashed red line) for the problem (Equation 93). On the left, ${\langle \pi_{0}\rangle }_{t}$; on the right, ${\langle \pi_{1}\rangle }_{t}$. In both panels, ω=0.5,γ=3, $\rho_{0}$ identified by the Bloch vector $\vec{a}=\left(0.2;\;0.3;-0.4\right).$ The curves overlap.

**Figure 4 entropy-27-00527-f004:**
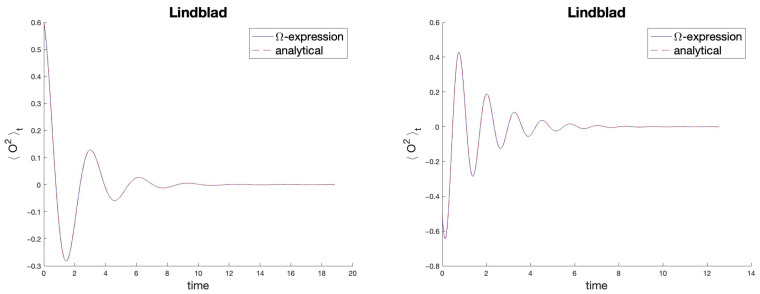
Comparison of the results offered by Expression (Equation 92) (solid blue line) with those known theoretically (dashed red line) for the problem (Equation 93) and the second-diagonal class $O^{2}$. On the left, c=1, d=1,ω=1,γ=1, $\rho_{0}$-Bloch vector $\vec{a}=\left(0.3,\;0.3,\;-0.4\right).$ On the right, c=−3,d=−1, ω=2.5,γ=1.3, $\vec{a}=\left(0.2,\;-0.1,\;0.6\right).$ The curves always overlap.

## Data Availability

The original contributions presented in this study are included in the article. Further inquiries can be directed to the corresponding authors.

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
