# Peer review of "Quantum Exact Response Theory Based on the Dissipation Function"

_entropy, 2025, doi:10.3390/e27050527_

Round 1

Reviewer 1 Report

Comments and Suggestions for Authors

The paper is interesting, door opening, and well written, but difficult to follow for someone not directly involved in this research.

Reviewer 2 Report

Comments and Suggestions for Authors

This is a very interesting paper in which the authors demonstrate clearly how exact response theory from classical statistical mechanics can be adapted for quantum systems. The theory is validated against various test cases, and is in excellent agreement with the expected results. The layout of the manuscript is logical, and the tests appear appropriate.

While there are many avenues which warrant further investigation, including the treatment of time-dependent Hamiltonians, infinite-dimensional Hilbert spaces, various aspects of open quantum systems, and the possibility of alternative forms of the dissipation operator (as the authors mention), this work lays a strong and comprehensive foundation while avoiding overcomplication. I therefore recommend that it should be accepted in its current form.

I have only some minor suggestions which the authors may wish to implement to improve clarity:

  • The Heaviside function appears to be omitted from the second to last equation in (63), and the font used for the quantity A appears inconsistent (unless there is some subtlety of notation with which I am not familiar).
  • The notation for the observables discussed in equations 96 and 98 appears a little inconsistent, with superscript numbers added in 100 to distinguish them. I would suggest including the appropriate superscripts from the beginning, and making them consistent with the axis labels in figure 4 which currently use subscripts.